

# Machine learning prediction of motor response after deep brain stimulation in Parkinson's disease—proof of principle in a retrospective cohort

Jeroen G.V. Habets[1], Marcus L.F. Janssen[1,2], Annelien A. Duits[3],
Laura C.J. Sijben[4], Anne E.P. Mulders[1,3], Bianca De Greef[4,5], Yasin Temel[1,6],
Mark L. Kuijf[4], Pieter L. Kubben[1,6,7] and Christian Herff[1]

[1] Department of Neurosurgery, School for Mental Health and Neuroscience, Maastricht University, Maastricht, The Netherlands
[2] Department of Clinical Neurophysiology, Maastricht University Medical Center, Maastricht, The Netherlands
[3] Department of Psychiatry and Neuropsychology, School for Mental Health and Neuroscience, Maastricht University Medical Center, Maastricht, The Netherlands
[4] Department of Neurology, Maastricht University Medical Center, Maastricht, The Netherlands
[5] Department of Clinical Epidemiology and Medical Technology Assessment (KEMTA), Maastricht University Medical Center, Maastricht, The Netherlands
[6] Department of Neurosurgery, Maastricht University Medical Center, Maastricht, The Netherlands
[7] Department of Neurosurgery, Radboud University Medical Center, Nijmegen, The Netherlands

Corresponding author
Jeroen G.V. Habets,
j.habets@maastrichtuniversity.nl

## ABSTRACT

**Introduction.** Despite careful patient selection for subthalamic nucleus deep brain stimulation (STN DBS), some Parkinson's disease patients show limited improvement of motor disability. Innovative predictive analysing methods hold potential to develop a tool for clinicians that reliably predicts individual postoperative motor response, by only regarding clinical preoperative variables. The main aim of preoperative prediction would be to improve preoperative patient counselling, expectation management, and postoperative patient satisfaction.

**Methods.** We developed a machine learning logistic regression prediction model which generates probabilities for experiencing weak motor response one year after surgery. The model analyses preoperative variables and is trained on 89 patients using a five-fold cross-validation. Imaging and neurophysiology data are left out intentionally to ensure usability in the preoperative clinical practice. Weak responders ($n = 30$) were defined as patients who fail to show clinically relevant improvement on Unified Parkinson Disease Rating Scale II, III or IV.

**Results.** The model predicts weak responders with an average area under the curve of the receiver operating characteristic of 0.79 (standard deviation: 0.08), a true positive rate of 0.80 and a false positive rate of 0.24, and a diagnostic accuracy of 78%. The reported influences of individual preoperative variables are useful for clinical interpretation of the model, but cannot been interpreted separately regardless of the other variables in the model.

**Conclusion.** The model's diagnostic accuracy confirms the utility of machine learning based motor response prediction based on clinical preoperative variables. After reproduction and validation in a larger and prospective cohort, this prediction model holds potential to support clinicians during preoperative patient counseling.

## INTRODUCTION

Subthalamic nucleus deep brain stimulation (STN DBS) is a widely accepted therapy for Parkinson's disease (PD) patients in which dopaminergic replacement therapy is unsatisfactory (*Deuschl et al., 2006*; *Limousin et al., 1995*; *Odekerken et al., 2013*; *Schuepbach et al., 2013*). In the majority of these patients, DBS can reduce motor symptoms or their fluctuations and thereby improve quality of life (*Williams et al., 2010*). Despite careful patient selection, some patients still show limited or no improvement of motor fluctuations and quality of life (*Williams et al., 2010*). Since the introduction of STN DBS, clinicians aimed to determine reliable predictors (*Pinter et al., 1999*).

Preoperative levodopa responsiveness of motor symptoms, severity of motor symptoms, and younger age are repeatedly reported as positive predictive factors for postoperative (Movement Disorders Society –) Unified Parkinson's Disease Rating Scale ((MDS-)UPDRS) motor improvement (*Kleiner-Fisman et al., 2006*). Contrarily, preoperative levodopa responsiveness is also reported to not predict STN DBS outcome (*Schuepbach et al., 2019*; *Zaidel et al., 2010*). Preoperative severe quality of life (QoL) impairment, more time spent in off-condition of dopaminergic medication, levodopa responsiveness, and low BMI are reported as positive predictive factors on postoperative QoL (*Abboud et al., 2017*; *Daniels et al., 2011*; *Frizon et al., 2018*; *Schuepbach et al., 2019*). Reports on the predictive value of disease duration, daily levodopa dosage, postural and gait impairment, and non-motor symptoms all show conflicting results (*Dafsari et al., 2018*; *Frizon et al., 2018*; *Kleiner-Fisman et al., 2006*; *Liu et al., 2018*). Comparison of reported motor outcome is hampered due to variance in assessment scales and assessments during varying dopaminergic states (*Goetz et al., 2008*).

These non-conclusive results maintain the need for a simple tool which neurologists can use in clinical practice to predict motor outcome after STN DBS for individual patients. To realize a usable and representative tool for the preoperative setting, our approach is limited to preoperative clinical variables. Preoperative prediction will always lack surgical information such as lead placement. This lack of information is inherent to any approach that aims to contribute to a better preoperative counselling.

Machine learning methods are increasingly used in medical practice to unravel patterns to improve understanding of clinical data (*Meyer et al., 2018*). Predictive machine learning models can be distinguished from traditional statistics by generating outcome predictions for new, individual patients, instead of correlations between pre- and postoperative variables on a group level. To ensure practical usability, clinical relevance, and interpretable results, the development and implementation of these models requires statistical, programming, and clinical expertise (*Kubben, Dumontier & Dekker, 2019*). To add value to PD care, predictive analysis should improve challenging clinical decision making instead of reproduce valid clinical decisions (*Ballarini et al., 2019*; *Cerasa, 2016*). Here, we report

the development and proof-of-concept of a prediction model that generates probabilities for weak and strong motor response one year after STN DBS for individual PD patients based on preoperative clinical variables.

## MATERIAL AND METHODS

### Study population

We considered patients who underwent STN DBS for PD in our academic neurosurgical centre between 2004 and March 2018. The surgical procedure is described in the Supplemental Material. We included 127 patients who completed one-year postoperative follow up during this period. We excluded patients who had missing UPDRS-III scores in their preoperative on-medication condition, or postoperative on-medication, on-stimulation condition.

The Medical Ethical Committee of Maastricht UMC+ approved this study (2018-0739). Informed consent was not obtained since the retrospective data was collected coded.

### Pre- and postoperative variables

All available preoperative demographic data, disease specific data (disease onset, disease duration, levodopa equivalent daily dosage (LEDD)) (*Esselink et al., 2004*), clinical performance scores ((MDS)-UPDRS, and Hoehn & Yahr (H&Y) scores), as well as relevant neuropsychological scores assessing executive functioning, in particular verbal fluency (semantic and lexical) and response inhibition (based on the interference score of the Stroop Colour Word Test) were incorporated. We left out imaging and neurophysiology data, to ensure the user-friendliness and accessibility in clinical practice during preoperative counselling. No analyses are required which ask software, hardware, or analysing knowledge.

All included preoperative clinical and neuropsychological scores were assessed in the on-medication condition and the available (MDS-)UPDRS III and H&Y scores in the off-medication condition were also included. Preoperative motor levodopa-responsiveness was calculated by subtracting UPDRS III scores in the off-medication condition with UPDRS III scores in the on-medication condition. Postoperative collected variables consist of UPDRS I, II, III and IV and H&Y scores in on-medication and on-stimulation conditions, UPDRS III in on-stimulation and off-medication conditions, and the performance on the verbal fluency and Stroop tests in on-stimulation and on-medication conditions. Both MDS-UPDRS and UPDRS scores were collected due to the variation in surgery dates among the population. To create uniform UPDRS scores, all MDS-UPDRS scores were recalculated to UPDRS scores (*Goetz et al., 2008*). Pre- and postoperative differences for UPDRS scores I until IV, H&Y scores, LEDD, and neuropsychological scores were calculated. Furthermore, we registered applied DBS voltage, frequency, and pulse width at one-year follow up. To compare DBS-settings, we computed the mean total electrical energy delivered (TEED) (*Koss et al., 2005*).

### Prediction model

The machine learning prediction model uses multivariate logistic regression analyses. This logistic regression model distinguishes itself from (univariate) correlative regression
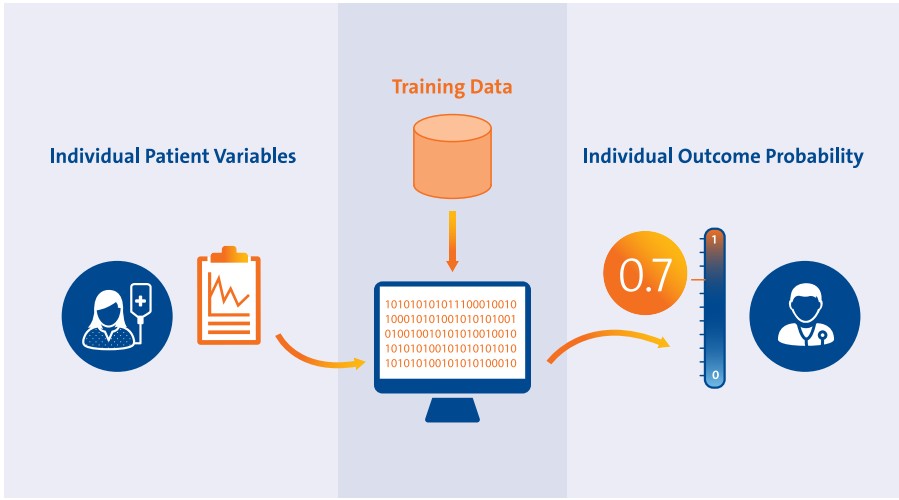

**Figure 1  Overview of prediction approach.** Workflow of the prediction model as a preoperative counselling tool. The preoperative individual patient variables are inserted in the prediction model, which is trained on the retrospective database ('Training Data'). The model calculates the probability to become a weak responder between 0 and 1, in this example 0.7. The clinician can use this probability to inform the patient during preoperative counselling.

models by generating outcome probabilities for individual patients (Fig. 1). We focused on motor response as outcome and differentiated between 'strong responders' and 'weak responders'. To capture a wide spectrum of motor responders, improvement on UPDRS II, III and IV was evaluated. A strong responder is defined as a patient who showed a minimal clinically important difference (MCID) on UPDRS II, III, or IV in on-medication and on-stimulation condition one-year postoperative vs. preoperative on-medication condition (see Fig. 2). MCID was defined as more than 3, 5, and 3 points improvement for UPDRS II, III and IV respectively, based on a literature review (see Supplemental Material). Patients who improved more than the MCID on UPDRS II or IV, but showed a deterioration on UPDRS III of more than the MCID of 5 points (*Horvath et al., 2015*), and the yearly natural disease progression of 2 points (*Holden et al., 2018*), together 7 points, were defined as weak responders (*Holden et al., 2018*; *Horvath et al., 2015*).

The prediction model uses the following available preoperative variables to generate an outcome probability: gender, age at DBS, PD duration at DBS, age at PD onset, UPDRS I, II, III and IV in on-medication condition, motor levodopa response, H&Y scale in on- and off-condition, the Stroop interference score, the verbal fluency scores, and the LEDD.

The logistic regression model was fitted, i.e., trained, on the relation between preoperative variables and postoperative outcome categorization (*Pedregosa et al., 2011*). We evaluated the trained model with a 5-fold cross-validation. This cross-validation fits, i.e., trains, the model on 80% of the patients, the 'training data'. During this 'training phase', a weight, '$\beta$', is assigned to every single preoperative variable, 'x'. The fitted model was then evaluated, i.e., tested, on the remaining 20% of patients in the database, the 'test data'. During this 'test phase', the preoperative variables of every individual patient in the test

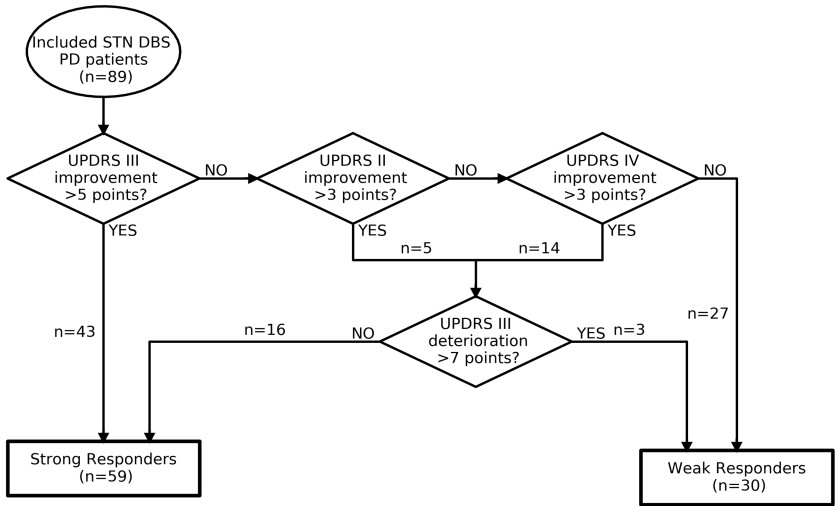

**Figure 2  Decision flowchart of outcome categorization.** DBS, deep brain stimulation; PD, Parkinson's disease; STN, subthalamic; UPDRS, Unified Parkinson Disease Rating Scale.

data were inserted in the model separately. The model generates an outcome probability to become a weak responder for every individual patient. This probability was generated by a calculation of all 'x' values of the inserted patient with the corresponding weights ($\beta$) using the logistic function $1 / (1 + \exp(-\beta * x))$. The generated probabilities from the test data are compared with the actual outcome to test predictive accuracy. The 5-fold cross-validation repeats these phases 5 times until every patient was used for testing exactly once. The cross-validation leads to less limitations in sample size regarding number of considered predictive variables (*Wynants et al., 2015*). Still, the small number of patients on which the trained model is tested during every iteration in this 5-fold cross-validation is a limitation of this approach. Evaluating the average performance over the 5 iterations gives the best assumption of the predictive performance of the model. We chose logistic regression as a prediction model instead of a deep learning-based model due to the relatively small database size and the fact that the weight, or influence, of every preoperative variable can be interpret easily. This interpretation helps to generate an intuition what the prediction is based on *Rudin (2019)*.

To use a certain prediction model in clinical practice, a threshold should be chosen to accept a probability. This means every probability above the threshold is regarded to be true (weak response in this model), and every probability below the threshold is regarded to be false (strong response in this model). The accuracy of the model is strongly dependent on the threshold. A common way to evaluate the overall performance of a prediction model is to plot the receiver operating characteristic (ROC). The ROC visualizes for different thresholds between 0 and 1 the corresponding true positive and false positive rates (Fig. 3A). Performance of prediction models is often expressed as the area under the curve (AUC) of the ROC (Fig. 3A). In clinical practice, a threshold should be selected before the model can be used as a prospective application.

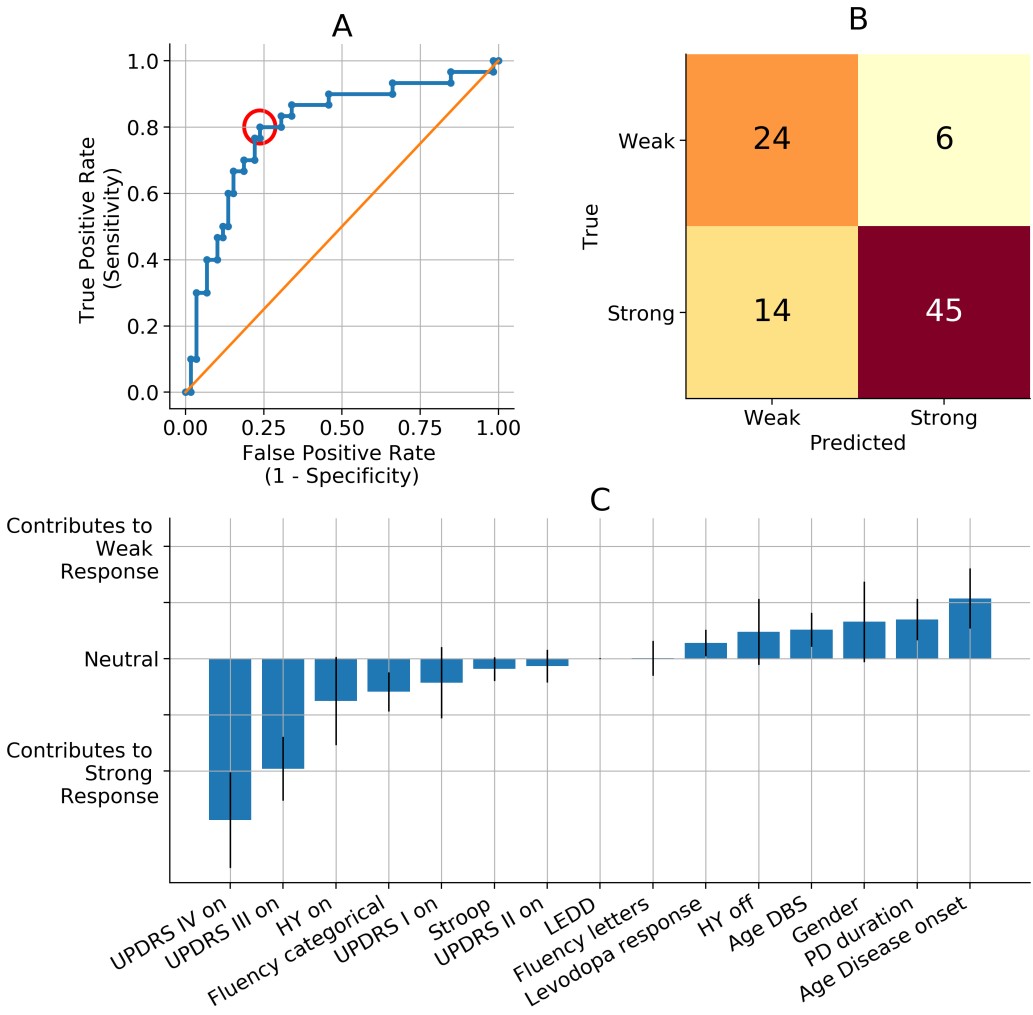

**Figure 3** **Prediction model performance and importance per predictive variable.** (A) Visualization of the performance of our prediction model. Our prediction model performs with an average area under the curve (AUC) of the receiver operating curve (ROC, blue line) of 0.78 (standard deviation: 0.08). All the dots on the ROC represent a threshold between 0 and 1 for accepting a probability to be a weak responder to be true. Every threshold leads to a different true positive rate and false positive rate. The red circle represents the threshold corresponding with B. The orange line represents chance level in which true positive rates equal true negative rates. (B) Confusion Matrix of the example when 0.29 is chosen as a threshold for accepting the probability to be a weak responder (red circle in A). The true positive rate of 0.80 results in 24 out of 30 true weak responders getting a true weak prediction. The false positive rate of 0.24 results in 14 out of 59 true strong responders getting a false weak prediction. The classification accuracy is 0.78 with 69 out of 89 correct predicted patients. (C) Relative influence of all preoperative predictive variables. The blue bars represent the normalized Odds Ratios. The heights represent the effect on prediction outcome of a 1 unit increase in the specific variable, while all other variables stay equal. AUC, area under the curve; DBS, deep brain stimulation; H&Y, Hoehn & Yahr scale; LEDD, levodopa equivalent daily dosage; Levodopa response, difference between UPDRS III off-medication minus UPDRS III on-medication; off, off-medication; on, on-medication; ROC, receiver operate characteristic; TEED, total electrical energy delivered; UPDRS, Unified Parkinson Disease Rating Scale; PD, Parkinson's disease.

To understand which variables are important in the prediction model, we can explore the importance of every separate preoperative variable. Variable importance is expressed

as 'weights'. To make these weights interpretable, they are converted to Odds Ratios by calculating $\exp(\beta)$, and normalized afterwards. These normalized Odds Ratios are called 'relative influences', and they denote the change in probability to be a weak responder when the respective variable increases 1 unit, and all other variables stay equal (Fig. 3A).

Comparative descriptive analysis between preoperative and postoperative variables and between weak and strong responders are performed with Mann–Whitney-U-tests.

To facilitate prediction models, we imputed missing data-points in preoperative variables. (For further explanation on the Random Forest imputations applied on preoperative variables, please see Supplementary Material). To prevent imputations of variables that are the target of prediction, we did not impute postoperative variables. Analysis is performed in Python Jupyter Notebook 3 (Jupyter Team, https://jupyter.org, revision fe7c2909) using packages pandas (version 1.0.4), Numpy (version 1.16.4), scikit-learn (version 0.21.2), and Scipy (version 1.3.0). We report our findings according to the TRIPOD Checklist for Prediction Model Development (*Collins et al., 2015*).

## RESULTS

### Preoperative and postoperative variables

We included 89 patients with a well-documented one year follow up after STN DBS, 37 patients were excluded due to missing data points in UPDRS III score in preoperative on-medication condition, or postoperative on-medication and on-stimulation condition. We report descriptive statistics containing the original data (no imputed preoperative data). The total group showed statistically significant postoperative improvements in UPDRS III scores, both compared with preoperative on- and off-medication conditions, and in UPDRS IV scores. We observed a significant decrease in LEDD (Table 1). Further, there was a significant deterioration in neuropsychological scores on a group level.

59 out of 89 patients were categorized as strong responders, 30 patients were categorized as weak responders (Fig. 2). Postoperative clinical records until one-year follow-up were evaluated and surgical factors explaining weak response were ruled out for all weak responders. The groups had significant differences on all postoperative UPDRS scores and differences, except for the UPDRS III during on-stimulation and off-medication state (see Table 2). We observed no significant or relevant differences between the groups regarding neuropsychological scores, LEDD, or TEED.

### Performance of the prediction model

The prediction model has a good general performance with an average AUC of the ROC of 0.79 (standard deviation: 0.08) (Fig. 3A).

When 0.29 is chosen as a threshold for accepting probabilities to become a weak responder, this leads to a true positive rate of 0.80 and a false positive rate of 0.24 (Figs. 3A–3B). This corresponds to a positive predictive value of 0.63 and a negative predictive value of 0.88. Selecting 0.29 as the threshold for probability acceptance leads to a classification accuracy of 78%, since 69 out of 89 patients are predicted correctly.

The relative influence values represent the influence, or weight, of each preoperative variable in the prediction model (Fig. 3C). Older age at PD onset has the strongest relative

**Table 1  Preoperative and postoperative variables of total population.**

| | Baseline characteristics[a] |
|---|---|
| Female sex | 37 (42%) |
| Age | 61 (8) |
| Disease duration | 10.7 (5.1) |
| Preoperative UPDRS III levodopa response | −18.6 (13.1) |
| Preoperative UPDRS III % levodopa response | −45.0 (38.0) |

| | Preoperative[a] | 1 year follow up[a] |
|---|---|---|
| UPDRS I[b] | 1.3 (1.3) | 3.0 (3.1) |
| UPDRS II[b] | 9.8 (6.6) | 9.6 (5.5) |
| UPDRS III[b] | 21.9 (12.5) | 16.4 (9.9)[e] |
| UPDRS III[c] | 39.1 (13.1) | 16.4 (9.9)[e] |
| UPDRS IV[b] | 5.5 (4.0) | 2.8 (2.4)[e] |
| H&Y 1[b] | 2 (2%)[d] | 4 (3%) |
| H&Y 1.5[b] | 2 (2%) | 1 (1%) |
| H&Y 2[b] | 13 (15%) | 21 (30%) |
| H&Y 2.5[b] | 34 (40%) | 24 (34%) |
| H&Y 3[b] | 25 (29%) | 19 (27%) |
| H&Y 4[b] | 9 (11%) | 2 (3%) |
| H&Y 5[b] | – | – |
| Fluency total categories[b] | 39.7 (9.4) | 33.6 (9.8)[e] |
| Fluency total letters[b] | 35.5 (10.8) | 33.6 (11.9)[e] |
| Stroop interference[b] | 56.1 (35.1) | 76.7 (63.1)[e] |
| LEDD (milligrams) | 1187 (619) | 656 (510)[e] |
| TEED | – | 134 (130) |

**Notes.**

H&Y, Hoehn & Yahr scale; LEDD, levodopa equivalent daily dosage; off-/on-med, off-/on-medication; off-/on-stim, off-/on-stimulation; TEED, total electrical energy delivered; UPDRS, Unified Parkinson Disease Rating Scale.

[a] Values are given as mean and standard deviation of the mean.

[b] Preoperative: on-medication, postoperative: on-medication and on-stimulation.

[c] Preoperative: off-medication, postoperative: off-medication and on-stimulation.

[d] Percentage of Hoehn and Yahr scales are relative based on the number of available data (pre: $n = 85$, post: $n = 71$)

[e] Significant difference with p-value $< 0.05$, calculated with Mann Whitney-U test.

influence for becoming a weak responder. High preoperative UPDRS III and IV scores in the on-medication condition are the strongest predictors for becoming a strong responder (Fig. 3C). Additionally, a high preoperative UPDRS II score, high scores on the categorical fluency and Stroop interference test, and higher H&Y score in the on-condition were moderate predictors for becoming a strong responder.

# DISCUSSION

## Proof of concept of machine learning prediction in preoperative DBS outcome counselling

The presented machine learning model differentiated between individual weak and strong motor responders one-year after STN DBS for PD with a good overall predictive performance, the AUC of the ROC was 0.78, and the classification accuracy was 0.78% (Figs.

**Table 2** Comparison of postoperative variables in groups with strong responders and weak responders.

| 1 year follow up variables | Strong responders, $n = 59$[a] | Weak responders, $n = 30$[a] |
|---|---|---|
| UPDRS I[b] | 2.2 (2.2) | 4.5 (4.0)[f] |
| UPDRS I change[c] | 0.4 (1.9) | 1.5 (1.7)[f] |
| UPDRS II[b] | 8.4 (4.7) | 12.3 (6.1)[f] |
| UPDRS II change[c] | −2.8 (6.3) | 5.6 (5.9)[f] |
| UPDRS III, on-med[b] | 13.9 (7.5) | 21.5 (11.9)[f] |
| UPDRS III change[c] | −11.9 (11.6) | 7.3 (8.5)[f] |
| UPDRS III, off-med[d] | 20.9 (13.1) | 25.7 (6.7) |
| UPDRS III change[e] | −29.0 (13.8) | −12.4 (14.4)[f] |
| UPDRS IV[b] | 2.4 (2.2) | 3.5 (2.6)[f] |
| UPDRS IV change[c] | −4.1 (3.6) | 0.1 (4.2)[f] |
| Fluency total categories[b] | 33.8 (10.4) | 33.4 (8.5) |
| Fluency total letters[b] | 31.2 (11.5) | 31.8 (12.8) |
| Stroop interference[b] | 75.6 (66.4) | 79.0 (55.7) |
| LEDD | 622 (511) | 717 (501) |
| LEDD change | −509 (472) | −577 (529) |
| LEDD change (%) | −40.7 (37.5) | −42.2 (35.2) |
| TEED | 145 (151) | 112 (69) |

Notes.

LEDD, levodopa equivalent daily dosage; off-/on-med, off-/on-medication; off-/on-stim, off-/on-stimulation; TEED, total electrical energy delivered; UPDRS, Unified Parkinson Disease Rating Scale.

[a] Mean (standard deviation)

[b] On-stimulation, on-medication at one-year follow up

[c] Difference between on-medication and on-stimulation vs. preoperative on-medication

[d] On-stimulation and off-medication at one-year follow up

[e] Difference between on-stimulation and off-medication vs. preoperative off-medication

[f] Significant difference with $p$-value < 0.05, calculated with Mann Whitney-U test

3A–3B). These results contribute to a proof-of-concept of machine learning prediction of individual postoperative motor outcome, solely based on preoperative clinical variables. We want to stress that these results are the first step towards the clinical utilization of smart supportive computational models in the delicate, multifactorial decision-making process of DBS therapy counselling. To increase the likelihood of creating a beneficial clinical impact for the patient, a model should be interpretable for clinicians, generalizable to the aimed patient population, and the effect of a utilization on the quality of clinical care should be investigated (*Kelly et al., 2019*).

## Interpretation of the predictive performance and the clinical utilization

A predictive machine learning model for clinical support generates individual outcome probabilities range from 0 to 1, rather than binary classes. The presented confusion matrix is an example of a clinical utilization where probabilities to become a weak responder higher than 0.29 were accepted (see Figs. 3A–3B). The selection of this threshold will eventually determine the model's clinical behaviour, usefulness, and its potential clinical impact on patient care. The value of this threshold leads to a different balance between false positive and false negative predictions (Fig. 3B), and should be validated on an external cohort. The

presented threshold is chosen to realize a good accuracy (78%) and to fit to the intended clinical utilization of this model. Since the majority of STN DBS candidates will experience a strong response, it is important that the clinician can trust a strong response prediction (negative predictive rate (0.88)). Also, the model should create awareness about the chance of becoming a weak responder in case of increased risk. This requires a good true positive rate, here 0.80.

Further, the confusion matrix shows that most incorrect predictions are actual strong responders who get a weak responder prediction. The final decision will be accurately guided by the experience of the DBS team and will overrule the majority of these predictive inaccuracies. Therefore, the actual clinical usefulness and impact should be investigated in a prospective clinical study. Moreover, these numbers and considerations emphasize that a clinical decision support tool in a precarious setting as preoperative counselling for DBS therapy should have a warning role, instead of a directive role. The goal should be to support the clinician with validated numerical expectations, and ensure her or his awareness in case of a patient with a higher than average chance on suboptimal therapeutic effect.

## The clinical value of predicting STN DBS motor response in the preoperative phase

Establishing an accurate prediction tool for motor outcome after STN DBS facilitates the clinician to improve patient counselling, expectation management, postoperative patient satisfaction, and potentially even patient selection (*Lin et al., 2019*). Due to the complexity and heterogeneity of individual STN DBS candidates, outcome prediction needs to be accompanied by a clinical expert's appraisal. Moreover, the accuracy of a prediction model solely regarding clinical preoperative factors will always be limited due to the influence of surgical factors. Nevertheless, we intentionally chose to leave pre-, intra- and postoperative imaging and neurophysiology variables out of our model. This way, we ensure the model's accessibility and usability in clinical practice. We aim to provide the clinician during preoperative counselling with numerical support regarding the most probable motor outcome for an individual patient.

Further it is important to underline this model's target patient population and clinical utilization. The model is designed for, and tested on, PD patients who were included for STN DBS implementation. This means the model should be applied to patients which are highly likely to be included for STN DBS implementation in the current care practices. In this population, the model is aimed to inform the clinician, and indirectly the patient, about a potential increased risk on a suboptimal motor response. This means the model is not developed to identify optimal STN DBS candidates from a general PD population.

## The additive value of machine learning methods for clinical decision support tools

The applied predictive multivariate logistic regression model was chosen to overcome limitations inherent to conventional (univariate) logistic regression models (*Daniels et al., 2011*; *Frizon et al., 2018*; *Schuepbach et al., 2019*). Traditional predictive or correlative analyses mainly result in a correlation between one preoperative variable and a postoperative

outcome, while controlling for several confounding preoperative variables. The absence of confounders and predictive variable selection in machine learning models, makes them less limited by sample size than traditional correlative analyses (*Wynants et al., 2015*). The presented prediction model distinguishes itself by evaluating all available variables simultaneously. The applied cross-validation decreases the restriction due to sample size and leads to less a-priori selection-bias. Nevertheless, the advantages of machine learning predictive models come with specific analysing risks. For example, an external validation is required to evaluate under- or overfitting of the model, and validation of the threshold for accepting probabilities. Further, we stress the importance of using interpretable predictive machine learning models. In contrast to more complicated models such as deep neural networks, interpretable machine learning models remain explainable. This is essential in evaluating clinical validity and creating clinical confidence in a supportive decision tool which are both important in realizing actual clinical impact (*Rudin, 2019*).

## Interpretation of the preoperative predictive variables in this model

This overview of interpretable weight of each predictive variable is an advantage of the applied logistic regression in the prediction model (Fig. 3C). This advantage enables clinicians to verify whether the ratio behind the predictions is clinically valid or whether predictions are based on unexpected variables.

The reported large influence of higher age at PD onset on becoming a weak responder is in agreement with a finding of a meta-analysis that report younger age to be a positive predictor for a favourable outcome. Contrarily, the same meta-analysis reports longer PD duration as a predictor for favourable outcome (*Kleiner-Fisman et al., 2006*).

Preoperative UPDRS III and IV scores in the on-medication condition have the largest relative influence values for becoming a strong responder in this model. High preoperative motor severity increasing the chance to become a strong responders is in line with the findings of a meta-analysis, although most included studies describe preoperative severity in the off-medication condition (*Kleiner-Fisman et al., 2006*). Evidence on the predictive value of symptom severity in on-medication condition is limited. The finding that H&Y scores do not majorly influence outcome probabilities is in line with previous literature. This literature describes that disease severity positively influences the chance on strong motor response, while axial and balance problems negatively influence this chance (*Kleiner-Fisman et al., 2006*). Since H&Y severity is based on both these factors, an inconclusive effect is expected.

Furthermore, there is literature on predictive or correlative variables and QoL outcome after STN DBS. We cannot compare these findings one-on-one with our findings. However, our holistic outcome classification aims to cover multiple aspects which influence QoL. High preoperative UPDRS III scores, and high UPDRS III levodopa response, are identified as important predictors of good QoL outcome, and motor outcome (*Daniels et al., 2011*; *Frizon et al., 2018*; *Kleiner-Fisman et al., 2006*). Conversely, recent studies have failed to replicate this positive predictive value of UPDRS III severity, or UPDRS III levodopa response on QoL outcome, or motor outcome (*Abboud et al., 2017*; *Schuepbach et al., 2019*; *Zaidel et al., 2010*). Our findings are in line with some of these findings, since the absolute

UPDRS III score showed a relevant influence, while the UPDRS III difference between on- vs. off-medication condition did not have a relevant influence. Regarding the reported influence of levodopa responsiveness, one should consider that LEDD is expressed in milligrams, which means that the relative influence of a unit increase (1 milligram) is not a clinically relevant increase.

A high score on categorical Fluency is a small contributor to becoming a strong responder (Fig. 3C). A high categorical Fluency score corresponds to better neuropsychological functioning. The contribution of the Stroop interference score is very small. Thus, there is no large influence of neuropsychological tests in our prediction model.

Our lack of QoL scores prevented replication of previous findings which suggested that an impaired preoperative QoL-functionality predicts a large postoperative QoL improvement (*Liu et al., 2018*; *Schuepbach et al., 2019*). Likewise, the absence of a proper non-motor symptom scale hampered potential reproduction of the recently described importance of non-motor symptoms (*Dafsari et al., 2018*).

The reported influences of the preoperative variables on the outcome probability are mainly consistent with the literature, and are partly contradicting literature. We stress that the reported influences of this model cannot be seen outside the scope of this model. They are only reported to gain insight in the underlying weights which determine the probabilities.

They cannot be interpreted on their own within individual patients when other variables in the model are disregarded.

## Limitations

Our study is limited by its retrospective character. Missing preoperative data points were overcome by imputations. Outcome values were not imputed to prevent training of the model based on imputed self-generated data. Even though the imputation method was sound, the imputed values will never reach true values and will influence outcomes. Second, the internal consensus on the applied categorization for motor outcome is based on scientific grounds, but can always be disputed. The holistic approach including UPDRS II, III and IV, aims to cover aspects of daily life activities, motor symptoms, and adverse effects of treatment. Future work should include QoL metrics and investigate the correlation between (QoL) and the presented classification. We argue our approach in the Supplementary Material. Lastly, the accuracy of a preoperative prediction model will always be limited and contain variance due to the lack of surgical variables.

## CONCLUSION

The presented prediction model identified strong vs. weak responders one-year after STN DBS for PD with a good classification accuracy. The potential distribution of predictive inaccuracies was in line with the aimed clinical utilization. These findings contribute to the proof-of-concept of machine learning prediction of individual motor outcome after STN DBS based on preoperative clinical variables.

The reported preoperative variables cannot be interpreted separately outside the scope of this prediction model, but endorse the clinical reliability of the applied method.

These results and considerations support the potential and the timely relevance of predictive clinical support tools for DBS outcome, and advocate further reproduction and validation in a representative, multicenter cohort. The optimal clinical utilization should be refined and the clinical additional value and impact should be clarified before a predictive clinical support tool can be applied during individual preoperative DBS counseling.

**Abbreviations**

| | |
|---|---|
| **BMI** | body mass index |
| **H&Y** | *Hoehn and Yahr scale* |
| **LEDD** | *levodopa equivalent daily dosage* |
| **(MDS) UPDRS** | *(Movement Disorders Society) Unified Parkinson's Disease Rating Scale* |
| **MR** | magnetic resonance |
| **PD** | Parkinson's disease |
| **QoL** | quality of life |
| **STN DBS** | subthalamic nucleus deep brain stimulation |
| **TEED** | total electrical energy delivered |

## ACKNOWLEDGEMENTS

We would like to thank Jackson Boonstra for proofreading the manuscript. We would like to thank Geertjan Zonneveld for technical assistance on the design of Fig. 1.

### Funding

This research did not receive any specific grant from funding agencies in the public, commercial, or not-for-profit sectors. Yasin Temel and Pieter Kubben received an unrestricted grant from the Weijerhorst foundation. The funders had no role in study design, data collection and analysis, decision to publish, or preparation of the manuscript.

### Grant Disclosures

The following grant information was disclosed by the authors:
Weijerhorst foundation.

### Competing Interests

The authors declare there are no competing interests.

### Author Contributions

- Jeroen G.V. Habets conceived and designed the experiments, performed the experiments, analyzed the data, prepared figures and/or tables, authored or reviewed drafts of the paper, and approved the final draft.
- Marcus L.F. Janssen conceived and designed the experiments, analyzed the data, authored or reviewed drafts of the paper, and approved the final draft.
- Annelien A. Duits conceived and designed the experiments, authored or reviewed drafts of the paper, and approved the final draft.

- Laura C.J. Sijben, Anne E.P. Mulders performed the experiments, authored or reviewed drafts of the paper, and approved the final draft.
- Bianca De Greef, Yasin Temel, Mark L. Kuijf and Pieter L. Kubben conceived and designed the experiments, authored or reviewed drafts of the paper, and approved the final draft.
- Christian Herff conceived and designed the experiments, performed the experiments, analyzed the data, prepared figures and/or tables, authored or reviewed drafts of the paper, and approved the final draft.

## Human Ethics

The following information was supplied relating to ethical approvals (i.e., approving body and any reference numbers):

The Medical Ethical Committee of Maastricht UMC+ approved this study (2018-0739).

## Data Availability

The raw data is available in the Supplemental Files and is also available at DataverseNL: Habets, Jeroen; Janssen, Marcus; Herff, Christian, 2020, ''Development Individual STN DBS prediction model for PD'', https://doi.org/10.34894/FUTGYT, DataverseNL, V2.

## Supplemental Information

Supplemental information for this article can be found online at http://dx.doi.org/10.7717/peerj.10317#supplemental-information.

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
