# Peer review of "Machine learning prediction of motor response after deep brain stimulation in Parkinson’s disease—proof of principle in a retrospective cohort"

_PeerJ, doi:10.7717/peerj.10317_

## Round 0.1 · original submission · Major Revisions

Reviewer 1 requests validating the results in an external sample and notes that other data sets are available. Please address this concern especially. Other criticisms primarily focus on analysis or presentation. Please address each concern of both reviewers in your response letter.

Reviewer 1 ·

Basic reporting

176 ‘can be interpret easily’: typo.

Figure 3B: please use different colours (lowerright corner is unreadable).

Table 1: legend is incomplete.

Experimental design

See below.

Validity of the findings

See below.

Additional comments

Habets and colleagues describe an interesting study on a machine learning based prediction model to model success after DBS surgery (or rather: failure of DBS surgery). Their use of an advanced modelling approach is commended, as well as their use of preoperative clinical features to enhance the generalizability. The authors rightly state that this model is a proof-of-concept and should not be interpreted as a ‘final model’ for clinical practice.
There are however, several limitations with regard to insufficient information on variable-importance, a small sample size which is (incorrectly) stated to be overcome through cross-validation, and lack of external validation whereas more cohorts with these data are available.

73. “… limited or no improvement of motor disability”: do you intend motor fluctuations or severity of motor symptoms?

100-102 “To add value… clinical decisions”: you correctly state that challenging cases should be improved rather than valid decisions. However, given that you intend to develop a generic model the case-mix you require should resemble the case-mix from the actual population in order to prevent overfitting / underfitting.

111-113 “We included all… on-stimulation condition”. How many patients were initially considered (i.e. underwent STN DBS in the given time period) and how many patients were excluded for the mentioned reasons?

114-115: Was this study approved in the absence of informed consent, or was formal evaluation waived given the retrospective nature?

117-123: Please provide a list with the exact tests that were incorporated in the model (‘categorical, and letter tests’ are relatively vaguely descripted).

150-151: The reason for labelling a patient as a ‘strong responder’ in case of MCID improvement in any of the three domains can be understood, however if a patient improves in one domain and deteriorates in the two others this label may be subject for debate.

172-173: Using this approach (i.e. cross-validation) the test-set remains relatively small and you run the risk of testing on outliers. I agree with the merits of this approach (definitely better than split-sample validation) but the limitations should be discussed as well.

173: ‘predictive variables or confounders’: please remove the word ‘confounders’, there are no confounders in prediction modelling (as opposed to causal research).

176 ‘can be interpret easily’: typo.

221 “63 out of 90 patients were categorized as strong responders”. See earlier comment, please consider at least reporting the MCID improvements per domain separately as well rather than joined together.

231 Why choose 0.24 as a threshold? Even if the argumentation holds true, are you confident that this threshold would hold up in a prospective study of large sample size and therefore should be utilized in clinical practice?

Discussion:
I do find the discussion to be somewhat overoptimistic of the actual performance. It states that in 78% of cases the predicted class would be correct, however there are more outcomes relevant to PD care than just the UPDRS-classes (e.g. quality-of-life, satisfaction with surgery etc.). Moreover, the sample-size is by far too small to make any definitive claims on its utility. Although the authors have stressed this in the final paragraphs of the discussion, I would appreciate some more elaboration on the 'next required steps' before this research can proceed to clinical implementation.

279-285: see previous comments: there is no such thing as confounding variables in prediction modelling.

286-289: with such a small sample size: overfitting is still an issue. I do agree that cross-validation circumvents this issue, but definitely not entirely and the statement that cross-validation ‘decreases restriction due to sample-size’ is untrue. Moreover, cross-validation certainly does not prevent selection-bias as this is inherent to your study-design. An option is to provide ROC curves per cross-validation run to assess the dispersity of results (although this may not be very visually attractive).
Why was external validation not performed? There are a lot more similar cohorts available with which to increase your sample size and overcome several of the limitations mentioned in your discussion.

A rather large limitation of this study is that this model is based only on those patients that have been selected for surgery. Cases on which there is ‘preoperative’ doubt may or may not have been rejected for surgery and therefore there is a large likelihood of those patients being underrepresented in the training-set, thereby reducing its validity in a clinical setting.

Figure 3B: please use different colours (lowerright corner is unreadable).

Figure 3C: please provide exact numbers rather than visualization.

Table 1: legend is incomplete.

Reviewer 2 ·

Basic reporting

The basic reporting is excellent:

The submission is ‘self-contained,’ represents an appropriate ‘unit of publication’, and includes all results relevant to the hypothesis.
Coherent bodies of work are not inappropriately subdivided merely to increase publication count.

Experimental design

The experimental design is conceptualised and conducted very well:

This is original primary research within the Aims and Scope of the journal. The Research question is clear, relevant, timely and meaningful. It is stated how research fills an identified knowledge gap.
The investigation is conducted rigorously and to a high technical standard. The research is conducted in conformity with the prevailing ethical standards in the field.
Methods described with great detail. The raw data is provided to replicate the results.

Validity of the findings

The validity of the findings is excellent:

The raw data is provided to replicate the results. The benefit to literature is clearly stated.
All underlying data have been provided; they are robust, statistically sound, and controlled.
Conclusions are well stated, linked to original research question & limited to supporting results.

Additional comments

In this is a proof-of-principle study in 90 patients with Parkinson’s disease Habets et al. implement a machine learning logistic regression prediction model of “weak” and “strong” response to bilateral subthalamic stimulation. The model input parameters include a wide range of preoperative variables. Not abiding to the “rule-of-ten” is feasible because the authors use this machine learning prediction with a 10-fold cross-validation. The model predicts weak responders with a good C-statistic (0.88) and has 78% diagnostic accuracy.
This is a timely study by an experienced group, the statistical approach is sound and the primary research question of preoperative prediction of postoperative motor DBS outcomes is of key clinical importance. Overall, the authors have done an excellent job in explaining the advantages of the implemented machine learning logistic regression prediction model as opposed to the traditional univariate approach implement in a number of earlier studies.
Minor points:
- In 4 patients the motor improvement was predicted to be strong but was observed in truth to be weak. It is understandable to include only preoperative clinical parameters for the model. However, the authors may add that STN targeting was confirmed (e.g. based on postoperative imaging or intraoperative electrophysiology – at least in these 4 patients) so that it is clear that there were no lead misplacements which could account for false predictions in these 4 patients.
- The preoperative motor response was quantified by subtracting UPDRS-III ON – UPDRS-III OFF. Other studies have often used a ratio and reported the % improvement. Could the authors explain why they preferred a subtraction?
- Line 329 and 330: the term neuropsychological functioning is very broad and may be specified (executive functions).
- Line 332: higher preoperative QoL-scores predict larger QoL postoperative improvement

---

## Round 0.2 · accepted · Accept

Thanks for responding to the reviewers' comments.

Reviewer 1 ·

Basic reporting

I have no further comments.

Experimental design

I have no further comments.

Validity of the findings

I have no further comments.

Additional comments

I have no further comments.

Reviewer 2 ·

Basic reporting

no comment

Experimental design

no comment

Validity of the findings

no comment

Additional comments

This is the first revision of a proof-of-principle study on machine learning logistic regression prediction model of “weak” and “strong” response to bilateral subthalamic stimulation in 90 patients with Parkinson’s disease. The authors did a good job in implementing the most reviewer comments and the manuscript has improved.